# Molecular Basis of CO_2_ Sensing in *Hyphantria cunea*

**DOI:** 10.3390/ijms25115987

**Published:** 2024-05-30

**Authors:** Jian Zhang, Shiwen Duan, Wenlong Wang, Duo Liu, Yinliang Wang

**Affiliations:** 1School of Life Sciences, Changchun Normal University, Changchun 130033, China; 2School of Life Sciences, Northeast Normal University, Changchun 130024, China

**Keywords:** CO_2_ sensing, fall webworm, oviposition, foraging, sexual dimorphism, gustatory receptor

## Abstract

Carbon dioxide (CO_2_) released by plants can serve as a cue for regulating insect behaviors. *Hyphantria cunea* is a widely distributed forestry pest that may use CO_2_ as a cue for foraging and oviposition. However, the molecular mechanism underlying its ability to sense CO_2_ has not been elucidated. Our initial study showed that CO_2_ is significantly attractive to *H. cunea* adults. Subsequently, 44 *H. cunea* gustatory receptors (*GRs*) were identified using transcriptome data, and 3 candidate CO_2_ receptors that are specifically expressed in the labial palps were identified. In vivo electrophysiological assays revealed that the labial palp is the primary organ for CO_2_ perception in *H. cunea*, which is similar to findings in other lepidopteran species. By using the *Xenopus* oocyte expression system, we showed that the *HcunGR1* and *HcunGR3* co-expressions produced a robust response to CO_2_, but *HcunGR2* had an inhibitory effect on CO_2_ perception. Finally, immunohistochemical staining revealed sexual dimorphism in the CO_2_-sensitive labial pit organ glomerulus (LPOG). Taken together, our results clarified the mechanism by which *H. cunea* sense CO_2_, laying the foundation for further investigations into the role of CO_2_ in the rapid spread of *H. cunea*.

## 1. Introduction

CO_2_, as a ubiquitous gas in the natural atmosphere, is produced by almost all organisms when they obtain energy via respiration. Insects possess the ability to detect subtle changes in the concentration of carbon dioxide (CO_2_) in their environment [1]. CO_2_ not only serves as a danger signal for insects, but in social insects such as honeybees, CO_2_ could also cause an imminent increase in the nest temperature; thus, they expel CO_2_ by flapping their wings to lower the temperature of their nest [2]. CO_2_ is also important in host-seeking and oviposition in insects; e.g., blood-sucking mosquitoes use CO_2_ as a cue to locate their vertebrate hosts [3,4]; in phytophagous insects, evidence has shown that CO_2_ can guide *Elasmopalpus lignosellus* larvae to locate the freshest parts of the plant [5], while *Manduca sexta* and *Cactoblastis cactorum* use CO_2_ to locate suitable places to lay eggs [6,7]. However, the CO_2_-sensing mechanism of most other lepidopteran pests, such as the major forestry pest *Hyphantria cunea*, remains unclear. *H. cunea* is a highly polyphagous and fertile pest; one female can lay up to 500 eggs with a hatch rate of over 95% [8], and can forage on more than 400 plant species, resulting in great damage to forest ecosystems, which is known as “smokeless fires” locally [9]. One of the predominant reasons for their rapid spread is their ability to choose suitable foraging and oviposition sites [10], suggesting that CO_2_ cues might contribute to their adaptation to the environment.

The CO_2_ sensing pathway in insects varies among different taxa. Two CO_2_ receptor genes were found in *Drosophila melanogaster*, and only *DmelGr21a* and *DmelGr63a* were co-expressed in response to CO_2_ [11,12], suggesting that a heterodimer is needed in *D. melanogaster* to carry out CO_2_ sensing. However, CO_2_ receptor homologs have not been identified in honeybees, ants, or blacklegged ticks [13], suggesting that these species utilize distinct mechanisms for CO_2_ sensing [14,15]. Three CO_2_ receptors have been found in lepidopterans, such as *Danaus Plexippus*, *Bombyx mori*, *Heliconius Melpomene,* and *Helicoverpa armigera* [16,17]. A functional study of *H. armigera* showed that although *HarmGr1*, *HarmGr3,* and *HarmGr2* were expressed in the same neuron of the labial palps, only the co-expression of *HarmGr1*+*HarmGr3* or *HarmGr1*+*HarmGr2*+*HarmGr3* resulted in a robust response to Sodium bicarbonate (NaHCO_3_) [18], indicating that *HarmGr1* and *HarmGr3* were necessary for CO_2_ perception in *H. armigera*, whereas the role of *HarmGr2* is still unclear. To date, the molecular CO_2_ sensing pathway in *H. cunea* is unclear, and whether the CO_2_ sensing mechanism in *H. cunea* is conserved among lepidopteran species remains unknown.

At the central nervous system (CNS) level, the location and projection modes of CO_2_-sensing glomeruli also vary among different insects. In *D. melanogaster*, CO_2_-sensitive neurons in the antennae project to a ventrally distributed glomerulus called the *V glomerulus* [19,20]; in *Aedes aegypti* and in several other mosquitos, CO_2_-sensitive neurons in the maxillary palps are linked to the dorsomedial glomerulus in the antennal lobe (AL) [21,22,23]. These two species both have an ipsilateral projection; i.e., CO_2_ neurons project to only one side of the AL, which is also called the “single side projection mode”. In Lepidoptera, the well-accepted hypothesis is that labial palp pit organs (LPO) are used for CO_2_ sensing [24], and LPO neurons project to a specific glomerulus located on the most ventral side of the antennal lobe; this type of glomerulus is referred to as the “labial pit organ glomerulus” (LPOG) [24,25]. The LPO neurons in lepidopterans project to both the ipsilateral and contralateral antennal lobes via bilateral projection. LPOG is considered a specific CO_2_ glomerulus, and no other olfactory neurons project to this glomerulus; therefore, an assessment of its projection mechanisms and olfactory glomerulus volume could provide further motivation to assess the importance of CO_2_ sensing in *H. cunea* through higher-level mechanisms. In conclusion, although the rapid spread of *H. cunea* may be related to herbivore-induced plant volatiles (HIPVs) such as pinene [26], its ability to sense CO_2_ also plays a major role. But to our knowledge, no studies have investigated the role of CO_2_ in the spread of *H. cunea* using molecular-level approaches. Our study will contribute to a better understanding of the important role that CO_2_ plays in the dispersal of *H. cunea*, which will provide a theoretical basis for pest management.

## 2. Results

### 2.1. Effect of CO_2_ on H. cunea Behavior

To verify the effect of CO_2_ on the behavior of female *H. cunea*, Y-tube olfactometer tests were conducted with 60 *H. cunea* at different CO_2_ concentrations. Compared with those of the negative control (pure air without CO_2_, Figure 1), 32 moths preferred 1% CO_2_. As the CO_2_ concentration increased, 34 moths were attracted to the side with CO_2_ at a 3% concentration, 37 moths were attracted to 5% CO_2_, 38 moths were attracted to 8% CO_2,_ and 41 moths were attracted to 10% CO_2_. These results showed that CO_2_ concentrations ranging from 8%-10% had an attractive effect on *H. cunea* adults; within this range, the attractive effect of CO_2_ increased with increasing CO_2_ concentration.

### 2.2. Identification and Homology Analysis of HcunGRs

To identify CO_2_ receptors in *H. cunea*, 44 candidate gustatory receptors (*GRs*) were identified from *H. cunea* transcriptome data; 24 *GRs* were full-length sequences, and 20 were truncated sequences. Blast results revealed that the similarity between *HcunGRs* and those of other species ranged from 35% to 91% (Appendix A). *HcunGR16* was the homolog of the fructose receptors of *D. melanogaster*, *B. mori*, *H. armigera*, and *H. melpomene* and was also hypothesized to be a candidate fructose receptor in *H. cunea* (Figure 2) [16]. *HcunGR4-10* and *HcunGR12-14* clustered together with the conserved sugar receptor *DmelGR64* [27], suggesting that these *GRs* have a sugar receptor (non-fructose) sensing role. *HcunGR44* is a candidate for bitter receptors in *H. cunea*. Most importantly, we found that *HcunGR1*, *DmelGR21a*, and *HarmGR1*; *HcunGR2* and *HarmGR2*; and *HcunGR3*, *DmelGR63a,* and *HarmGR3* were grouped together, indicating that these three *HcunGRs* might play a role in CO_2_ sensing in *H. cunea*.

Notably, the expression levels of the *GRs* showed that *HcunGR1*, *HcunGR2,* and *HcunGR3* were highly abundant in the labial palps (Figure 3a). The expression patterns of *HcunGR1*, *HcunGR2,* and *HcunGR3* were further verified by q-PCR (Figure 3b), and the qPCR results were consistent with the heatmap. These results indicate that the main organ responsible for CO_2_ sensing in *H. cunea* is the labial palp, which is similar to that in other lepidopteran species.

### 2.3. Electrophysiological Response of the Antennae and Labial Palp to CO_2_

The results of electrolabialpalpography (ELPG) and electroantennogram (EAG) showed that the response to CO_2_ was significantly higher in the labial palp of female *H. cunea* than in males (Figure 4a) (*p* = 0.0313), but there was no significant difference in the antennae response to CO_2_ (Figure 4b). In addition, the response of the labial palp to CO_2_ was consistently higher than that of the antennae at 1%-10% concentrations (Figure 4). These findings suggest that the labial palp is the main organ involved in CO_2_ sensing in *H. cunea* and that female moths have stronger CO_2_-sensing abilities than males.

### 2.4. Two-Electrode Voltage Clamp (TEVC) Response of HcunGr1, HcunGr2, HcunGr3, and Their Combinations

By analyzing the dissolved CO_2_ concentration in NaHCO_3_ solution, a direct correlation between the concentration of NaHCO_3_ and the dissolved CO_2_ concentration was identified (Appendix A). Moreover, we found that Na^+^ in NaCl solution also elicits a channel current (Figure 5a–g and Appendix A). Therefore, eliminating the Na^+^ effect from the NaCl solution is necessary when calculating the “real response”; we used full response minus NaCl response to look at the real response of CO_2_ [28]. Oocytes expressing *HcunGR1*, *HcunGR2*, or *HcunGR3* alone or with co-expressions of *HcunGR1*+*HcunGR2* or *HcunGR2*+*HcunGR3* did not respond to CO_2_ after excluding the effect of Na^+^ (Figure 5c–g). However, the *HcunGR1+HcunGR3* and *HcunGR1*+*HcunGR2*+*HcunGR3* expression sets significantly responded to CO_2_ beginning at a concentration of 100 mM (equivalent to 51 ± 3 ppm CO_2_) (Appendix A), and the response increased gradually with increasing CO_2_ concentration (Figure 5a,b). In the range of 100–300 mM, the response of *HcunGR1*+*HcunGR3* ranged from 257 ± 87 nA to 1387.67 ± 162.7 nA, and the response of *HcunGR1*+*HcunGR2*+*HcunGR3* ranged from 245.67 ± 114.33 nA to 575.3 ± 89.3 nA (Figure 5h). The response of the *HcunGR1*+*HcunGR2*+*HcunGR3* set was significantly lower than that of the *HcunGR1*+*HcunGR3* set from a concentration of 200 mM or greater (Figure 5i).

### 2.5. Anterograde Dye Filling of Labial Palps in H. cunea

After we concluded that the labial palp is the main organ for sensing CO_2_, an anterograde dye-filling experiment on the labial palps was conducted to further explore the transmission of CO_2_ signals to the central nervous system. We observed the dye’s entry from the labial palp nerves, passing through the gnathal ganglion (GNG), then dividing into two bundles, and finally arriving at the LPOG (Figure 6a–f). The LPOG is located in the ventral region of the ALs, where it is similar to the DP region in *D. melanogaster*; we named this region DP1. The projections displayed a clear boundary, and the neurons exhibited a bilateral projection pattern.

To precisely locate and calculate the volume of the LPOG in the AL, we, based on the nine individuals (five females and four males), performed two-dimensional and three-dimensional reconstructions of all the glomeruli (Figure 6g–j) (Appendix A). It was found that female *H. cunea* possess 81 glomeruli, whereas male moths have only 74; most of the glomeruli were roughly spherical, while a few were irregularly shaped, forming a central fiber nucleus. After computing the surface area and volume of DP1 in both male and female *H. cunea*, we found that the total surface area of DP1 in females (5741.25 μm^2^) was nearly the same as that in males (5728.94 μm^2^); however, the average volume of DP1 in females (34,388.3 μm^3^) was significantly (*p* = 0.0115) greater than that in males (28,852.4 μm^3^) (Figure 6k), suggesting that sexual dimorphism existed in the DP1 glomerulus.

## 3. Discussion

Initially, we discovered that *H. cunea* adults exhibit a preference for CO_2_ concentrations ranging from 1% to 10%; this preference might be linked to the foraging and oviposition behavior of *H. cunea*. Studies have shown that many plants, such as Nepenthes, release in excess of 5% concentration of CO_2_ [29], which is consistent with the CO_2_ concentration in our behavioral test. In addition, researchers have found that adults *H. cunea* prefer to forage and oviposition at night [30], which is the peak time of CO_2_ release, which partially explains why the high concentration of CO_2_ could also attract *H. cunea* females. Previous studies have shown that the level of respiratory metabolism in plants is a main indicator reflecting plant quality. A high respiration rate indicates the presence of more nutrients, such as carbohydrates [7]; therefore, *H. cunea* can choose strong nectar as food by sensing the CO_2_ released by plants [31]. On the other hand, it has been shown that insect spawning increases with increasing CO_2_ concentration [6], suggesting that *H. cunea* can also choose oviposition sites by CO_2_ cues to maximize the survival of their offspring. Field observations have also shown that *H. cunea* prefer to lay eggs on the dorsal sides of leaves [32], which may be because the undersides of terrestrial plant leaves have more stomata and can generate higher CO_2_ concentrations [33]. *H. cunea* can use this CO_2_ cue to lay eggs on the dorsal side of the leaf to prevent damage from direct sunlight. In summary, our behavioral results showed that CO_2_ can attract adult *H. cunea*, which may be related to their oviposition behavior and its adaptation. And the possibility of further study about CO_2_ baited traps such as mosquitos will be an exciting topic in the future [34].

Homology analysis revealed that three CO_2_ receptor homologs exist in *H. cunea,* which is consistent with findings in other lepidopteran insects [16,17], indicating that the CO_2_ sensing pathway might be relatively conserved in Lepidoptera. The expression levels of the three candidate CO_2_ receptors demonstrated that they were all specifically expressed in labial palps; however, their expression was low in the antennae. And the response of the labial palp is significantly greater than that of the antennae, suggesting that the labial palp is the primary organ for sensing CO_2_ in *H. cunea*, as is the response of other lepidopteran species, such as *H. armigera* and *M. sexta* [18,25]. The fact that females produce stronger action potentials than males in the labial palp also suggests that CO_2_ may play an important role in female-specific behaviors such as spawn selection in *H. cunea*. In *D. melanogaster,* the molecular mechanisms for CO_2_ sensing in taste and olfaction are mutually independent [35], and ionotropic receptors (IRs) are involved in the detection of CO_2_ [36,37,38]. In addition, the response of *D. melanogaster* to CO_2_ is regulated by two different neural pathways, one for CO_2_ attraction and the other for avoidance [39]. Therefore, it is necessary to further investigate whether there are other CO_2_-sensing pathways in *H. cunea*.

In vitro expression of *HcunGR1, HcunGR2,* and *HcunGR3* revealed that CO_2_ responses occurred only when *HcunGR1* was combined with *HcunGR3* or when *HcunGR1* was combined with *HcunGR2+HcunGR3.* These results support the crucial role of the *HcunGR1+HcunGR3* co-expression in CO_2_ sensing in *H. cunea,* which is consistent with the findings in *H. armigera* [18]. Notably, the response of the *HcunGR1*+*HcunGR2*+*HcunGR3* ternary complex was significantly lower than that of the co-expression at concentrations greater than 200 mm. The role of *HcunGR2* in CO_2_ sensing remains unknown. Our results could be due to two possible explanations. One possibility is that the limited number of ion channels on the oocyte surface results in a reduction in the expression ratio of the *HcunGR1*+*HcunGR3* co-expression when *HcunGR2* is expressed [40], therefore suppressing the TEVC response of *HcunGR1*+*HcunGR3*; another possibility is that *HcunGR2* may serve as a modulator, playing an inhibitory role in the CO_2_ sensing process in *H. cunea*. In conclusion, we have shown that the co-expression formed by *HcunGR1*+*HcunGR3* plays a primary role in CO_2_ sensing, but the function of *HcunGR2* requires further exploration.

At the CNS level, a bilateral projection pattern of labial palps was observed in *H. cunea*; this result is consistent with what has been observed in *H. armigera* and *Anopheles gambiae* (*A. gambiae*) but differs from the unilateral projection pattern observed in *D. melanogaster* and *A. aegypti*. Our heatmap result showed that *GR1*, *GR2,* and *GR3* were the top three highest expressing receptors in labial palps, which were responsible for CO_2_ sensing; thus, the projection from the labial palp into the brain was mainly for CO_2_ sensing. But we also found that there are other GRs also expressed in the labial palp, which may be related to wider sensing of different cues [41]. And previous research has suggested that bilateral projections enable olfactory receptor neurons (ORNs) to release an asymmetric amount of neurotransmitters on both sides of ALs [42], which leads to a stronger signal in one projection neuron (PN) than in the other, enhancing the contrast of odor concentration gradients between the two brain hemispheres [43]. This asymmetrical projection pattern may help *H. cunea* detect the differences in CO_2_ concentrations between the two labial palps and rapidly locate the CO_2_ source. Moreover, the average volume of DP1 in females was significantly higher than that of males; however, the total surface area of DP1 was nearly the same in both genders. This indicates that the shape of DP1 is different between males and females, which is consistent with our observations that the male’s DP1 had an irregularly shaped (Appendix A). And the sexual dimorphism was observed in the morphology of LPOGs; the average volume of LPOGs was significantly greater in females than in males. It is well accepted that a larger glomerulus indicates greater odor sensitivity due to the greater number of synaptic connections [44,45]. An enlarged LPOG may reflect the crucial role of CO_2_ sensing in *H. cunea* females; in contrast, no significant sexual dimorphism was observed on the LPOG in *M. sexta* [46]. In summary, the identified sexual dimorphism of labial palp projections in *H. cunea* may somewhat explain the differences in CO_2_ function between males and females, which might be linked to female-specific behavior such as oviposition.

## 4. Materials and Methods

### 4.1. Insect

A single fall webworm egg mass was collected from a Manchurian ash (*Fraxinus mandschurica*) tree in Animal and Plant Park, Jinlin Province, China (43°86′96.71″ N, 125°33′29.82″ E), and was reared in an artificial incubator (BIC-300 artificial incubator, Boxun, Shanghai, China) at 26 °C, 80% RH and a 19:5 light: dark cycle beginning in 2019. After each 12 generations, the wild population was collected again from the same place and crossed with the laboratory colony for rejuvenation for more than 30 generations. The larvae were fed on the leaves of mulberry trees (*Morus alba*), and the adults were given a 10% sucrose solution for energy supplementation.

### 4.2. Binary Choice Assay

The airflow of CO_2_ (Juyang Company, Changchun, China) was controlled by a flow meter and mixed with Zero Air (21% O_2_ and 79% N_2_, CO_2_ free, Juyang Company, China) to ensure the delivery of 1%, 3%, 5%, 8%, and 10% CO_2_. A Y-tube olfactometer (1.6 cm in diameter, 7.2 cm in base and arm length) was used for the binary choice assay, and pure air without any CO_2_ was used as a negative control. At 7 p.m. (peak spawning), day 3 after emergence, females were selected. A moth was first placed at the entrance of the main arm, and a “choice” was recorded when it entered an arm and stayed for more than 30 s. If no choice was made within 5 min, the data were recorded as “no choice”. A total of 60 *H. cunea* were tested. The Y-tube was cleaned with hexane, and the position of the stimulus was exchanged before and after each test. All the assays were performed in a dark room with red light (Intelligent LED solutions, Berkshire, UK) to avoid light interference.

### 4.3. Homology Analysis of Gustatory Receptors (GRs)

The *GR* sequence of *H. cunea* used in this study was obtained from our previous transcriptome studies (Appendix A) [47], and the genome sequences of six lepidopteran species, *B. mori* (GCA_026075555.1), *M*. *sexta* (GCA_014839805.1), *H. melpomene* (GCA_900068175.1), *Pieris rapae* (GCA_905147795.1), *H. armigera* (GCA_026262555.1), and *D. plexippus* (GCA_009731565.1) were used for homology analysis, and *D. melanogaster* (GCA_000001215.4) and *A. gambiae* (GCA_000005575.1) were used as outgroups. The amino acid sequences of the *GRs* were first aligned by MUSCLE, after which Molecular Evolutionary Genetics Analysis (MEGA; State College, PA, USA) version 6 was used to construct a maximum likelihood (ML) tree with the Jones–Taylor–Thornton (JTT) model [48]. Bootstrap support values were based on 1000 replicates. The resulting homological tree was visualized with FigTree 1.42 (http://tree.bio.ed.ac.uk/software/figtree/, accessed 29 May 2023). And the fragments per kilobase of transcript per million fragments mapped (FPKM) were used to measure gene expression [49]. R (R Foundation for Statistical Computer, Vienna, Austria) version 4.1.3 was used to construct the heatmaps.

### 4.4. Electrophysiological Recording

EAG and ELPG were used to detect the electrophysiological responses of the antennae and labial palp to CO_2_ in *H. cunea*. Three days after emergence, females and males were selected at night (oviposition peak time). Glass electrodes were pulled by using a micropipette PC-10 (Narishige, Tokyo, Japan) and then filled with 1 M potassium chloride containing 1% polyvinylpyrrolidone. The reference electrode was subsequently inserted into one eye of the insects, while the recording electrode was placed in contact with the tips of the antennae and labial palp using the micromanipulator MP-12 (Syntech, Kirchzarten, Germany). The obtained signals were amplified by a high-impedance ac/dc preamplifier (Syntech, Kirchzarten, Germany). CO_2_ stimuli ranging from 1% to 10% were injected into a carbon-filtered and humidified airflow for 0.2 s to deliver the stimulus to the antenna and labial palp at 500 mL/min generated by an air stimulus controller CS-55 (Syntech, Kirchzarten, Germany). A minimum of 3 individuals were tested, and 3 puffs were performed for each antenna or labial palps. The EAG and ELPG data were acquired with EAG Pro version 2.0 software (Syntech, Kirchzarten, Germany) and normalized by the response of negative control (21% O_2_ and 79% N_2_, CO_2_ free) (Juyang Company, China) by the equation “relative EAG response = EAG response of CO_2_ / EAG response of negative control”. The data were subsequently analyzed with GraphPad Prism 6.0 (GraphPad Software, San Diego, CA, USA).

### 4.5. RNA Extraction, Expression Pattern Analysis, Quantitative PCR, and Cloning

Previous studies have shown that some GRs are differently expressed in the olfactory sensing organ (antennae) between males and females, but in non-olfactory organs (such as the head, chest, abdomen, and legs), no sex-based different expressions were detected; thus, we use mix samples of these body parts as a negative control [50]. The head, thorax, abdomen, leg, and labial palp carefully dissected from 15 individuals (female:male = 1:1) with DEPC-treated forceps under a stereomicroscope (Motic, Hong Kong, China). Then, female and male antennae were dissected from 15 individuals in the same way. Total RNA was isolated from homogenized body parts with TRIzol reagent (Invitrogen, Carlsbad, CA, USA) following the manufacturer’s protocol. After extraction, the total RNA concentration was assessed with a NanoDrop 2000 spectrophotometer (Thermo Fisher Scientific, Waltham, MA, USA) and 1% agarose gel (Sangon Biotech, Shanghai, China) electrophoresis.

For the qPCR assay, 1 μg of total RNA was transcribed into cDNA by using TransScript One-Step gDNA Removal and cDNA Synthesis SuperMix (TransGen Biotech, Beijing, China). qPCR was performed with a LightCycler 480 II Detection System (Roche, Shanghai, China) and TransStar Tip Top Green qPCR Supermix (TransGen Biotech, Beijing, China) under the following conditions: 94 °C for 30 s; 45 cycles of 94 °C for 5 s, 55 °C for 15 s, and 72 °C for 10 s; the β-actin gene was used as an internal control. The primers used in this study are listed in Appendix A. The qPCR results were analyzed via the 2^−ΔΔCT^ method [51]. The data were subsequently analyzed with GraphPad Prism 6.0 (GraphPad Software, CA, USA).

The primer was designed by Primer 3 (https://bioinfo.ut.ee/primer3-0.4.0, accessed on 29 June 2023) (Appendix A), the full ORF of GRs containing 5′UTR and 3′UTR were obtained by PCR and ligated to pUCm-T Vector (Sangon Biotech, Shanghai, China) for sequencing. Subsequently, the ORF of GRs were amplified by a specific primer and subcloned to pGEMHE using pEASY-Uni Seamless Cloning and Assembly Kit (TransGen Biotech, Beijing, China) with BamHI and HindIII restriction sites (New England Biolabs, Ipswich, MA, USA). The recombinant plasmids were transformed in DH5α (TransGen Biotech, Beijing, China) competent cell, and then plasmids were extracted with a SanPrep Column Plasmid Mini-Preps Kit (Sangon Bio, Shanghai, China). After transformation, the inserts were verified via DNA sequencing (Sangon Biotech, Shanghai, China).

### 4.6. cRNA Synthesis and Oocyte Microinjection

The full-length open reading frames (ORFs) of *HcunGr1*, *HcunGr2*, and *HcunGr3* were expressed in *Xenopus laevis* oocytes individually or in combination; thus, seven sets of oocytes were obtained expressing the following: *HcunGr1, HcunGr2*, *HcunGr3*, *HcunGr1*+*HcunGr2*, *HcunGr1*+*HcunGr3*, *HcunGr2*+*HcunGr3,* and *HcunGr1*+*HcunGr2*+*HarmGr3*. cRNAs of *HcunGr1*, *HcunGr2,* and *HcunGr3* containing the 3′ (126bp) and 5′ (43bp) Xenopus globin UTR from the pGEMHE vector were synthesized using the mMACHINE T7 Transcription Kit (Ambion, Austin, USA) according to the manufacturer’s instructions. RNA concentration and purity were analyzed by a NanoDrop 2000 spectrophotometer (Thermo Fisher Scientific, Waltham, USA) and 1% agarose gel (Sangon Biotech, Shanghai, China) electrophoresis. A total of 46 nL (36.8 ng) of relevant cRNA was microinjected into each oocyte at a 1:1 or 1:1:1 ratio by a NanoLiter 2000 injector (World Precision Instruments, Sarasota, FL, USA). Afterward, the injected oocytes were incubated at 18 °C for 2 to 8 days in Barth’s solution (96 mM NaCl, 2 mM KCL, 5 mM MgCl_2_, 0.8 mM CaCl_2_ and 5 mM HEPES; pH adjusted to 7.6 by NaOH) supplemented with 10 μg/ mL gentamycin, 50 μg/ mL tetracycline, 100 μg/ mL streptomycin and 500 μg/ mL sodium pyruvate.

### 4.7. CO_2_ Quantification and TEVC

Since a TEVC requires a liquid environment, it is not possible to directly measure the response of *GRs* to CO_2_ in air; we chose to quantify the function of *GRs* by their response to dissolved CO_2_ in NaHCO_3_ solution, which was previously described by Xu et al. [28]. In brief, the concentration of dissolved CO_2_ in NaHCO_3_ solution can be calculated by the following equation; thus, we can obtain different concentrations of dissolved CO_2_ by controlling the pH and the concentration of bicarbonate.
$$\left[CO_{2}\left(aq\right)\right]=\frac{10^{\mathrm{pKoverall}-\mathrm{PH}}\times [HCO_{3}^{-}{]}_{nominal}}{1+10^{\mathrm{pKoverall}-\mathrm{PH}}}$$

K_overall_, which is sometimes referred to as K_a_, is a constant that incorporates the CO_2_ hydrolysis constant (Kh) and the first dissociation constant of carbonic acid (K_a1_), i.e., K_overall_ = K_h_ × K_a1_. The pKa value is 6.3 (https://pubchem.ncbi.nlm.nih.gov/compound/sodium-bicarbonate#section=pKa, accessed on 6 August 2023).

The TEVC technique was used to record the channel currents in *Xenopus* oocytes at a holding potential of −80 Mv [52]. Signals were amplified with an Axonclamp 900A amplifier (Molecular Devices, San Jose, CA, USA) and 50-Hz low-pass filters and digitized at 1 kHz. Data acquisition and analysis were performed using Axon Digi 1550B and pCLAMP10 software (Molecular Devices, San Jose, CA, USA). We used the same concentration of Na^+^ in NaCl solution as negative control and then used full response minus NaCl response to look at the real response of CO_2_. The data were subsequently analyzed with GraphPad Prism 6.0 (GraphPad Software, San Diego, CA, USA).

### 4.8. Anterograde Dye Filling and Immunohistochemical Staining of the Labial Palps

In order to furthermore explore labial palps projection on AL, the anterograde dye filling was performed on labial palp with both genders. According to previous studies [53], the adult insects were fixed in a plastic tube with dental wax so that their heads were exposed. The base of the labial palp was then cut off, and the fluorescent dye, tetramethylrhodamine dextran (MicroRuby, Molecular Probes; Invitrogen, Eugene, OR, USA), was applied at the cutting surface by using a needle. After staining, the insects were placed in a refrigerator with moist filter paper overnight, allowing transportation of the dye to the sensory axons. The next day, the brain and ventral nerve cord were dissected in Ringer’s saline, fixed in 4% paraformaldehyde (PFA) for 1 h, dehydrated via an EtOH gradient, cleared with methyl salicylate, and mounted in Permount on perforated aluminum slides with coverslips.

For immunohistochemical staining, the brains of *H. cunea* were carefully removed with forceps, fixed with 4% PFA at 4 °C overnight, and then rinsed with PBST three times for 45 min. The fixed brain body part was then transferred to 5% normal goat serum (NGS; Thermo Fisher Scientific, Waltham, MA, USA) and preincubated at 4 °C for 15 h. After preincubation, the primary antibody 3C11 (anti-SYNORF1, 1:100 dilution with 5% NGS and PBST) (DSHB, University of Iowa, Johnson County, IA, USA) was applied and incubated at 4 °C for 5 days. Afterward, the brain was rinsed with PBST again and treated with the secondary antibody Cy2 coupled to an Alexa FluorTM 488 (1:300 dilution with 1% NGS and PBST) (Invitrogen, Eugene, OR, USA) at 4 °C for 3 days. Finally, the brain was dehydrated using an alcohol gradient, cleared with methyl salicylate, stored at 4 °C, and mounted with 1 mm aluminum slides for confocal laser microscopy imaging.

The slides were observed under a laser scanning confocal microscope (LSM880, Carl Zeiss, Jena, Germany) with an excitation wavelength of 488 nm and collected between 490 nm and 560 nm. A clear image was captured with ZEN v2.6. The identified neuropils within the brain and ventral nerve cords were reconstructed by using the 3D reconstruction software Amira 5.4.3 (FEI, Hillsboro, OR, USA) [54].

### 4.9. Statistical Analysis

One-way ANOVA followed by Tukey’s test or Dunnett’s test was used for multigroup comparisons (Figure 3b). Wilcoxon signed-rank test was used as a comparison between two curves (Figure 4). When two sets of data were compared, the data were first evaluated by the Shapiro–Wilk normality test. If *p* ≤ 0.05, the data sets will be applied for the Mann–Whitney test. If *p* > 0.05, the data will subsequently apply to the F-test to check the equal variances. If the data pass the F-test, it will apply to unpaired *t*-tests (Figure 6k and Appendix A). If they do not pass the F-test, the data will apply to the unpaired *t*-tests with Welch’s correction. The data were analyzed with SPSS 27.0 (IBM Corp., Armonk, NY, USA) and GraphPad Prism 6.0 (GraphPad Software, San Diego, CA, USA). And for electrophysiological and TEVC experiments, the value of each biological replicate is the average of its three technical replicates, and three technical replicates mean that when conducting an antenna and labial palp or oocyte preparation, the same stimulus was applied three times.

## 5. Conclusions

Although the detailed biological functions of CO_2_ in foraging and oviposition in *H. cunea* could not be fully clarified, some conclusions were reached at this stage. First, CO_2_ (ranging from 1% to 10%) strongly affects *H. cunea* adults; second, the main organ involved in CO_2_ sensing in *H. cunea* is the labial palp, and female moths have a more sensitive ability to perceive CO_2_ than male moths, whereas *HcunGR1* and *HcunGR3* are indispensable elements in the CO_2_ sensing process; and third, sexual dimorphism is observed in the volume of the LPOG, the main CO_2_ projection region in the antennal lobe. In summary, these results showed that CO_2_ has an attractive effect on *H. cunea*, which may be related to their oviposition behaviors; by identifying the CO_2_ receptors in *H. cunea*, the olfactory sensing pathway was further clarified. These results would benefit the development of a CO_2_-based trap for the monitoring and control of *H. cunea*, meanwhile providing more evidence on the biological function of CO_2_ among varied insects, especially those pests that take serious damage to the agriculture and forest.

## Figures and Tables

**Figure 1 ijms-25-05987-f001:**
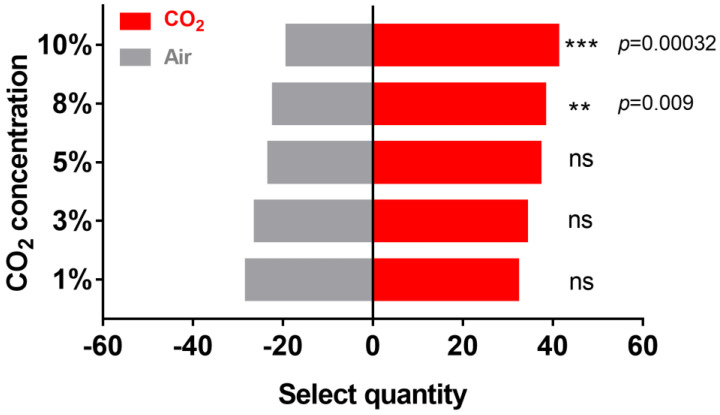
The number of female *Hyphantria cunea* attracted to different concentrations of CO_2_. Statistical differences were evaluated via the Chi−square test. ** *p* < 0.01, *** *p* < 0.001, ns: no significance.

**Figure 2 ijms-25-05987-f002:**
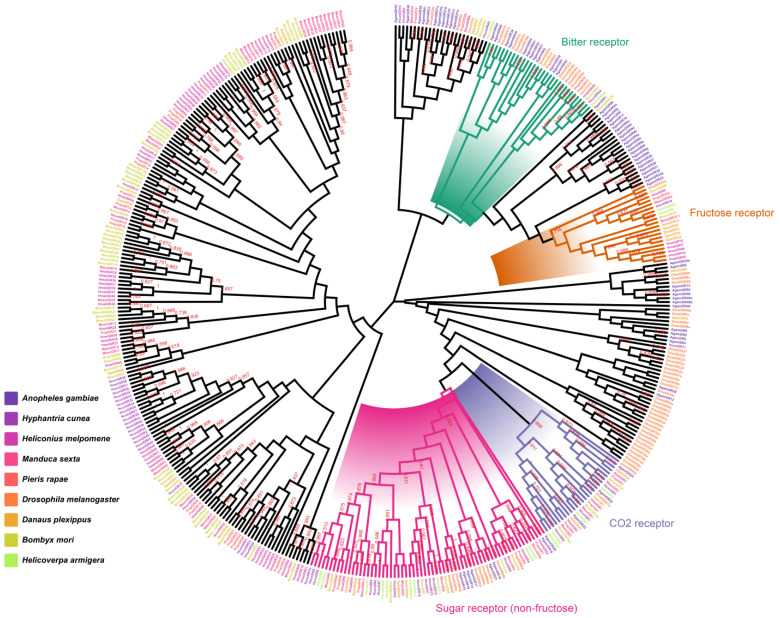
The maximum likelihood tree of candidate gustatory receptors (*GRs*). Bootstrap replications up to 1000.

**Figure 3 ijms-25-05987-f003:**
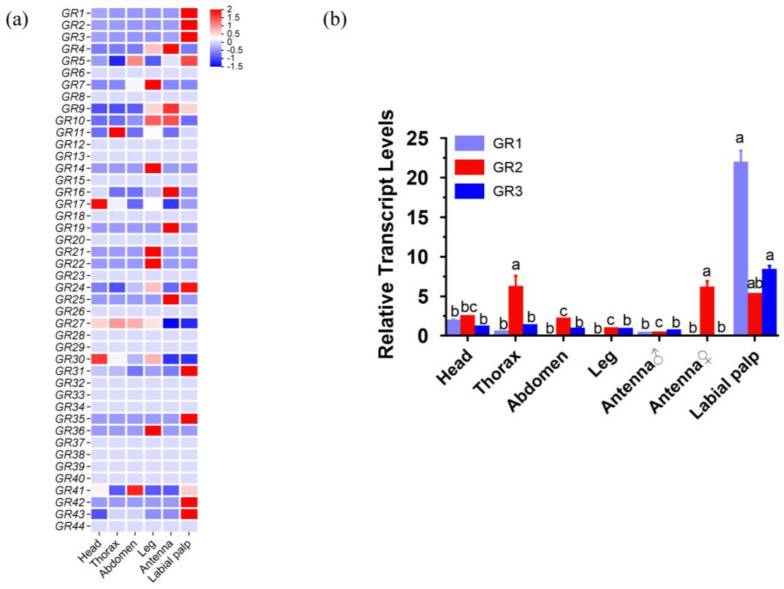
Expression profiles of *HcunGRs* in different body parts of adult *H. cunea*. (**a**) Characteristic expression patterns of 44 *HcunGRs* in different body parts based on FPKM (normalization by row). FPKM: Fragments per kilobase of transcript per million fragments mapped. (**b**) Expression patterns of three candidate CO_2_ *GRs* in different body parts of adult *H. cunea*. A different lowercase indicates a significant difference based on one-way ANOVA followed by Tukey’s multiple comparison test (*p* < 0.05). The data are presented as the means ± standard errors of the means (SEMs), *N* = 3.

**Figure 4 ijms-25-05987-f004:**
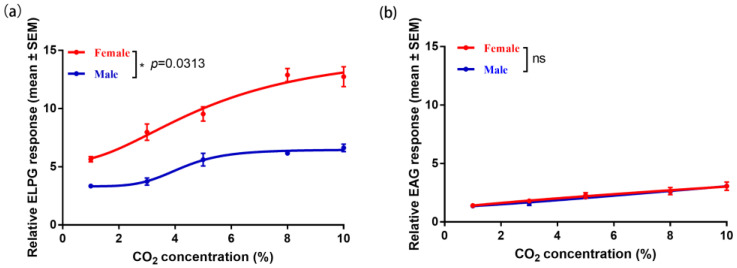
Electrolabialpalpography (ELPG) and electroantennogram (EAG) nonlinear regression curve for CO_2_. (**a**) Response of the labial palps of the *H. cunea* to CO_2_; (**b**) response of the antennae of the *H. cunea* to CO_2_. Statistical differences were evaluated by the Wilcoxon signed-rank test. * *p* < 0.05, ns: no significance. The data are presented as the means ± SEMs, *N* = 3 biological replicates (Appendix A).

**Figure 5 ijms-25-05987-f005:**
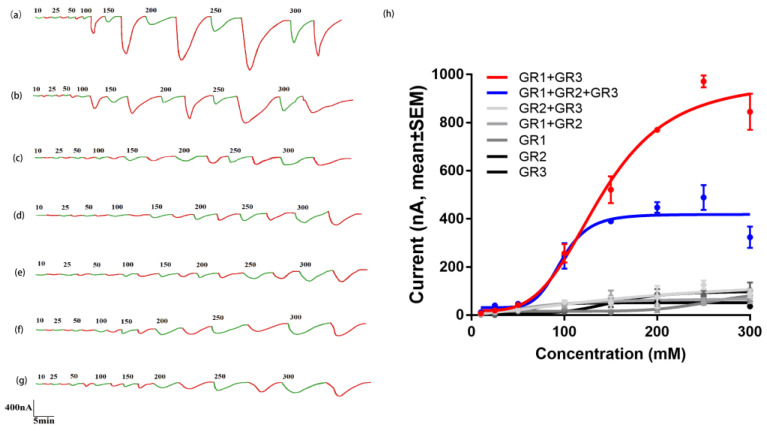
Two-electrode voltage clamp recording (TEVC) responses of *HcunGR1, HcunGR2,* and *HcunGR3* alone and in combination with different concentrations of NaCl and NaHCO_3_. (**a**) Response of *HcunGR1+HcunGR3*; (**b**) response of *HcunGR1+HcunGR2+HcunGR3*; (**c**) response of *HcunGR1*; (**d**) response of *HcunGR2*; (**e**) response of *HcunGR3*; (**f**) response of *HcunGR1+HcunGR2*; (**g**) response of *HcunGR2+HcunGR3*; green traces represent the response in NaCl solution; red traces represent the response in NaHCO_3_ solution; the number involved in (**a**–**g**) indicates concentration NaCl and NaHCO_3_; (**h**) nonlinear regression curve of *HcunGR1, HcunGR2,* and *HcunGR3* alone and in combination after excluding the influence of Na^+^. *N* = 3 biological replicates (Appendix A); The data are presented as the mean ± SEM.

**Figure 6 ijms-25-05987-f006:**
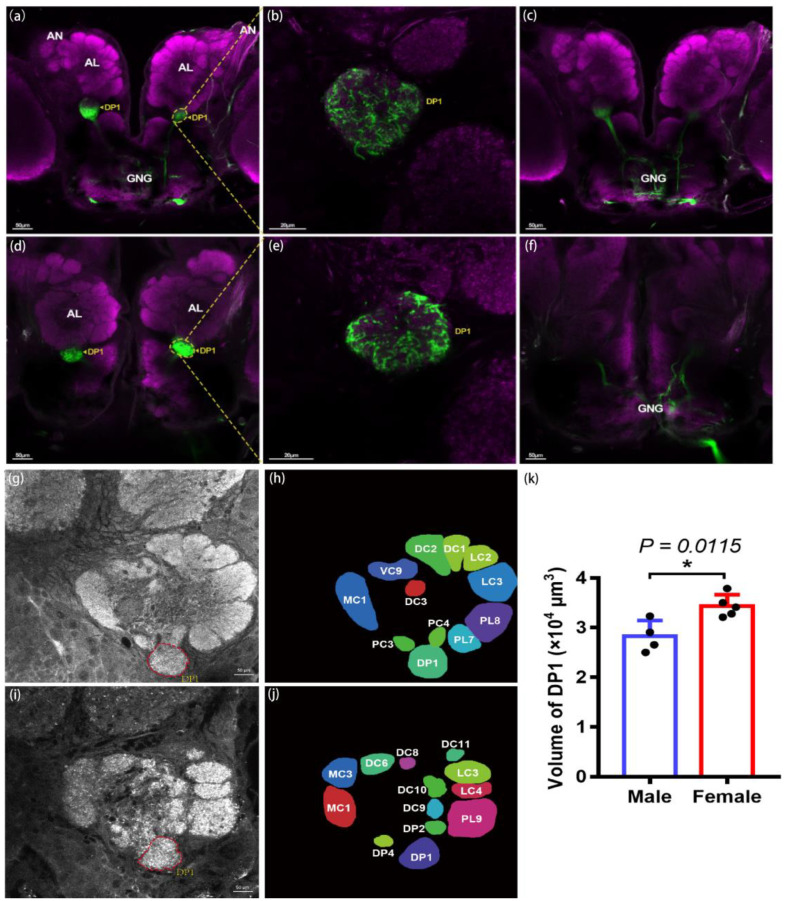
Anterograde dye-filling of labial palps and two-dimensional reconstructions of the antennal lobe (AL) in the DP1 region of *H. cunea*. (**a**–**c**) The central projections of female labial pit organ sensory neurons passing through the gnathal ganglion (GNG) to DP1; (**d**–**f**) the central projections of male labial pit organ sensory neurons passing through the gnathal ganglion (GNG) to DP1; (**g**,**h**) confocal images of the male *H. cunea* AL glomeruli seen from the ventral view. The sections are from anterior to posterior at a depth of 156 μm. Scale bar = 50 μm; (**i**,**j**) confocal images of female *H. cunea* AL glomeruli taken from the ventral view. The sections are from anterior to posterior at a depth of 142 μm. Scale bar = 50 μm; (**k**) the volume compares male and female DP1s. The data are presented as the mean ± SEM, *N* ≥ 4. Statistical differences were evaluated by unpaired *t* tests. * *p* < 0.05.

## Data Availability

The data sets analyzed in the current study are available from the corresponding author upon reasonable request.

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
