# Peer review of "Molecular Basis of CO_2_ Sensing in *Hyphantria cunea"

_ijms, 2024, doi:10.3390/ijms25115987_

Round 1

Reviewer 1 Report

Comments and Suggestions for Authors

Comments:

1. Lines 38-39 Is there any evidence that the pest uses CO2 or any relationship between CO2 concentration and suitable oviposition site?

2. Lines 70-71 The authors wrote that LPOG can specifically response to CO2. But why the ability to sense CO2 is inextricably linked to the rapid spread? why not other information, such as pinene, etc.

3. Line 134 What is 'multiple t tests'?

4. Lines 198-200 I cann't understand. Does H. cunea adult feed nectar? and is it evidence that H. cunea lays eggs near flower?

5. Lines 205-206 I think the evidence be the basis of present study.

6. Lines 220-221 Have you found H. cunea's avoidance behavior of high CO2 concentration?  The figure 1 does not support this.

7. Line 225 Why the authors thought that the HcunGRq and HcunGR3 make a heterodimer?  Also Line 234, a heterodimer has not been physically proved.

8. Lines 249-255 It is not easy to understand the logic. Please rewrite the sentences.

Reviewer 2 Report

Comments and Suggestions for Authors

In this paper, Zhang et al. analyze CO2 sensing of the forest pest Hyphantria cunea performing behavioral, electrophysiological, molecular and neurophysiological studies. In specific, they tested CO2 preference by behavioral assays in Y-tube olfactometer, despite not distinguishing between males and females; they tested CO2-chemosensory response by EAG as well as electrical labial palp gram (ELG), despite not distinguishing males from females; they performed expression analysis (FPKM and qRT-PCR) of GR-receptors that they identified in a previous transcriptomic study and characterized by a phyloenetic analysis, despite not distinguishing males from females in all body parts; they tested CO2-activation via GR1/2/3 expression or co-expression in oocytes from Xenopus; they backfilled the gustatory palp nerve to identify a specific AL-glomerulus that they name DP1, as a candidate region of the AL processing for CO2 sensing from labial palps.

My main concerns from this manuscript are the following:

1.) I don't see a precise distinction of results from males and females, in terms of behavior and insect electrophysiology, especially in the frame of the importance of the results coming from the comparison of the surface and volume of the DP1 between the two genders. I am convinced that at the light of the evidence that females have a bigger DP1 than males, there would have been much more interesting discussions if behavioral and electrophysiological findings shed light on differences between the sexes. A similar concern of lack of gender-distinction can be rose from the expression analysis, where there is no distinction between males and females among the various insect body parts that were tested for GR-expression.

2.) I have difficulties to understand how can be possible that backfilling a whole nerve from the labial palp organ, it will result in only one AL-glomerulus being positive to tetramethylrhodamine dextran. Taking this as true (I am not expert on this technique), I am not convinced that the result of backfilling may necessary relate only CO2 sensing, but rather a wider sensing of different cues from all the other neurons present on the labial palp.

3.) The methods are insufficiently described and I see a total lack of description on the statistics, which is also incomplete. In addition, I think that it is important to provide supplementary tables reporting raw data from electrophysiiology (on both live insects and Xenopus).

4.) Isn't it necessary to provide an accurate ethical statement for the use of Xenopus? How did you collect oocytes? What about your institutional permissions?

Said that, despite I think that this study is interesting and it should be published, unless it does not solve the aforementieond concerns, it is not ready for publication.

Please find below a list of my comments and the PDF attached, to help the authors with their improvements.

Line 87, figure 1 caption: Males or females? Larval or adults?

Line 88: Figure 1 caption: Paired or two samples T-test? Same for the other captions.

Line 90: 2.2. I think that it is better to provide also the corresponding accession numbers of the HcunGRs, based on Table S1. Otherwise it is too difficult to figure out which are which by just watching at Figure 2.

Line 107: figure 2 caption: Bootstraps values are not convincing. I see that some nodes have white colors, assuming that they have a value of 0. Please replace this way to indicate bootstraps in numbers.

Figure 3 (a and b): I think that it is important to indicate also for heads, thoraxes, abdomens, legs and labial palps, whether they belong to males or females. It is not precise as it actually is.

Line 118: Figure 3 caption. Indicating body parts as tissue is extremely wrong. I notice that this is a wide pattern from various authors nowadays, I don't know why. But these have to be call body parts, because they are made of various organs and tissues. From now on, please replace "tissue(s)" with "Body part(s)" everywhere is convenient.

Line 126: please use "Electroantennogram" to mean EAG.

Line 134: statistical difference between the responses from antennae and palps? Please specify. T-test: homo or heteroscedastic (did you compare variances of your data)? And before doing a T-test, did you test if your data are normally distributed? This info should be well specified in the methods.

Line 142: note for figure S2: please use the same order between the bar chart on the left (a) and the voltage clamp responses shown on the right (b-i). Maybe it's easier if you swift GR1+GR2 with GR1+GR3 in a.

Lines 142-143: how did you eliminate the Na+ effect? It is not described in the methods.

Line 147: 100 mM refers to the concentration of NaHCO3 and not to CO2 concentration. Please provide the exact CO2 concentration +- SEM based on your expected aq CO2 from Figure S1.

Lines 149-150: please indicate SEM.

Line 152: I would replace this "at" with a "from".

Line 156: what do the numbers above the TEVC indicate? Please specify in the caption. Please specify here that they are the tested concentration of NAHCO3 (as long as they seem to be).

Line 164: like line 134.

Line 167: As in my concerns described above, "to further explore...nervous system". I am not expert in this technique in particular, but I suppose that by backfill staining the nerve of the labial palp, brain regions processing signals from different kind of gustatory stimuli coming from the labial palp should be tracked, and not only the CO2 ones. In this frame, I am surprised that this backfilled staining tracks only one glomerulus (LPOG = DP1). However, apart from that, I think that it is important to stress that DP1 activation can be related with CO2 sensing but not CO2 only, given the fact that the results show backtrack of the whole palp nerve. 

Line 176: Figure S2 doesn't show the reconstruction of the AL-glomeruli. Please remove.

Lines 179-180: the average surface area of the DP1 glomerulus, is it referring to the widest section of it?

Line 185: Caption of figure 6:  It is not mentioned in the caption what you indicate in the figure at c and f as GNG (gathal ganglion). I think that it is good to mention (c,f) as evidence of the dye passing from the labial palp nerves, passing through the gnathal ganglion (GNG), then dividing into two bundles, and finally arriving at the LP1.

Lines 202-204 and reference 28: are you sure? I had a look at Clark 2019, "Leaves of terrestrial plants generally have more stomata on their undersides. Floating leaves of aquatic plants have stomata only on their upper surfaces, while some underwater plants have no stomata at all. Ad this is obvious, to prevent leaves dehydration.

Line 217: I rather suggest to add the following references, which are more targeted to the topic of IR-mediated CO2-sensing: - Ai et al. 2010 https://doi.org/10.1038/nature09537 - Ai et al. 2013 https://doi.org/10.1523/Jneurosci.5419-12.2013.

Line 271: twenty adults. Males or females?

Line 278: "from our previous transcriptome study" which one? Please cite it.

Lines 288-290: I think that this section deserves more details or at least citations to previous publications/projects of the authors when it comes to the identification of the FPKM values that have been shown in this research.

Line 320: the 2-ddct method, you should cite Livak and Schmittel 2001 -doi:10.1006/meth.2001.1262.

Lines 323-324: Please specify what these primers span exaclty. I do not see ATG and all the forwards and stop codons on all the reverse primers, you should mention if 5´upstram and 3´UTR are part of your amplification, as you mention it below.

Lines 328-330: you should give more details about cloning in E.coli.

Lines 336-337: how long the UTRs were exactly?

Line 339: how did you measure the RNA concentration?

Line 349: "Pingxi", shouldn't it be Xu et al. ?

Line 335: what does Kh refere to?

Line 364: "Mass staining was performed." the sentence like this is confusing. Could you incorporate it in a longer one?

Line 371: which alcohol? MeOH?

Comments on the Quality of English Language

English needs some minor adjustments. For the rest it's fine.

Reviewer 3 Report

Comments and Suggestions for Authors

I am not proficient in molecular genetics so I cannot evaluate that part of the paper

METHODS/RESULTS

-Choice test

There is always CO2 in the environment, and this CO2 level varies from day to day and depending on the number of people in the room and things like that. The authors do not report the use a CO2 detector to measure CO2 levels. This is a serious flaw of the study and they should acknowledge it. Just indicateng a hiher or lower % of the CO2 tank in the air tank is a very rough way of indicating the quantity of CO2. Please indicate this shortcoming.  

Twenty adults were used. So then, it is not clear how they obtained the error bars? What is the mean of 20 individuals in a test of proportions? This needs to be explained.  

Please indicate age of the individuals, sex, time of the insects day/night when the test was performed, light intensity level, characteristics of the red light (lux, wavelength, if available)

-Extracellular electrophysiology

I would favor ELPG over ELG to avoid confusions in future works

It is not a good idea to place the tip recording electrode on the tip of the palp as it blocks the opening through which the CO2 enters.

Please indicate that the labial palp of some moths has mechano-gustatory sensilla and that some of the response could be due to their stimulation (https://doi.org/10.1038/s41598-022-21825-w)

Please indicate age of individuals and the time of their day/night cycle when the tests were performed.

L296 1% polyvinylpyrrolidone: Plese explain the function of this compound

L296: "Inserted in the eyes",: in both eyes??

L302: "each individual" How many were tested total? Where the EAG individuals also palp-tested?

L304. Please indicate what is exactly the "negative control" and how it was normalized

Did I see any description of the statistical analysis? I think not.

RESULTS

-Figure 1. Choice test. I am not convinced. There are error bars but the methodology says that only a handful of insects were tested and there is no reference to groups from which a mean can be obtained (and it is not really needed since the result it is a ratio), and the significance to a 11:9 or 8:12 seems rather unlikely. There is something fishy about this test and it needs to be re-explained, reanalyzed or re-done.

-Figure 5 EAG/ELPG. It would be a lot better to do a dose-response curve (package drc in R) and calculate the slope and the ED50.

Not clear how these statistical tests were performed. What is compared with what?

Figure caption: "N = 3 biological replicates, 1 biological 135 replicate = 3 technical replicates". What do they mean? It is not clear at all.

Figure caption: "Electrical antenna gram" is incorrect

Y axis label: mV (relative response, mean ± SEM).

DISCUSSION

In moths the labial palps also serve for taste (Amat paper). It is still unclear which of the projections from the LP into the brain are for CO2 and which for taste (https://doi.org/10.1038/s41598-022-21825-w)

L171. The LPOG is dorsal? Please check your images and the literature

L179-183. It is interesting  that male and female glomeruli have the same surface but different volume. This is related to differences in shape. And also to the small sample size. Please discuss a little bit this point?

L214. I strongly doubt that the antenna has any function on CO2 sensing. The tiny responses in this study do not support it.

L219-221. The authors did not find any evidence of CO2 aoidance with high CO2 concentrations. Please give relevance to your results. I do not see any evolutionary advantage for an adult moth to be repelled by high CO2

L237. Again, remember that the labial palps are involved in taste, and the bilateral innervation could be related to taste and not to CO2 (https://doi.org/10.1038/s41598-022-21825-w)

 L249 (Female moths use their labial palps for searching 249 for oviposition sites based on CO2 cues) Please provide references

Comments on the Quality of English Language

Choice test and EAG/ELPG: they need to improve the presentation of the methodology and statistics, and data presentation

They are missing some recent literature

Reviewer 4 Report

Comments and Suggestions for Authors

The present article by Zhang et al on the molecular mechanism of CO2 perception in moth H. cunea is well designed and reasonably well presented. The article is reasonably well written, but there are some areas that needs some modification that I note in more detail below. Before I go further, one of the main issues in the article is throughout any words are clumped together without any space between them! Please go through them carefully and rectify them as I am not going to point out those here and they are numerous. Below are the list of areas that could be improved in order to make the article more relevant and readable to varied groups of people.

1.        Line 43: CO2 homologues = CO2 receptor homologues.

2.        Line 50: NaHCO3 = first time use of abbreviation without expansion.

3.        Either in intro or in discussion I believe it would be useful to have some theoretical discussion (with some references-if available) as to how CO2 could be used for pest management. While it sure could be used, but by reading the article it is not clear to the readers how that could be achieved.

4.        Line 84: is there any concentration beyond which CO2 would be aversive to the moths?

5.        Fig. 1: label y-axis

6.        Line 99: are candidate bitter = are candidate for bitter

7.        Section 2.2 in results and Fig. 2: some of the things that you discuss w.r.t. homology of different receptors could be better represented in figure if you could have zoom in panels for those branches from the main fig. 

8.        Section 2.5 and its corresponding section in M&M: you are performing anterograde dye filling (a term more utilized in the field) here (https://www.sciencedirect.com/science/article/pii/S1467803901000160), please refer it to as such and not “mass staining”.

9.        Line 176: Fig. S2 does not represent what you are referring to here.

10.  By reading your discussion, it is not clear how CO2 influences oviposition behavior in H. cunea. Some reference could be added w.r.t. that if any work has been done in that direction. In the same area, how does the amount of CO2 released by plants that are host to the moth correspond to the concentrations used here in the paper? Having some of that information in the discussion would make your article more relevant and could at possible ways it could be used in pest management. The main reason I ask this is CO2 is not expelled by plants at the same rate throughout the day and increases at night when there is no sunlight for photosynthesis.

11.  Line 271-272: it is not clear from this section if the moths were tested individually or in groups. Please clarify that.

12.  Line 302: indicate how many individuals were tested/recorded in this section.

13.  Section 4.8: was anterograde dye filling and antibody labeling done on same set of brains? It is not clear by reading this section.

14.  Line 364: the line “mass staining was performed” could be deleted.

15.  Line 394: occurs in = is observed in.

16.  Line 396: molecular = could be deleted. Just “revealed the mechanism…” should be good.

Comments on the Quality of English Language

The English language is ok, but please refer to my comments above regarding this for more details. Also, I again indicate any specific changes in the comments above.

Round 2

Reviewer 2 Report

Comments and Suggestions for Authors

I notice improvements in this version of the manuscript, but unfortunately, it is not ready for publication yet. For this reason, I think that it deserves further revisions.

As I said during my previous iteration, this research is interesting and it deserves to be publish but I am convinced that it is still incomplete. However, if the authors will address my suggestions below, I thinkt that they have reasons to attempt resubmission to this journal.

Up to now, I am convinced that if addressing my comments, the manuscript will be ready, or at least definitely improved for one last iteration.

Major concerns:

1.) Introduction: I think that the introduction is not properly structured. There are a lot of examples but the reader cannot catch what the authors want to say. I think that it is better to give start with a wider explaination of the context beween lines 26 and 27 " Insects possess the ability to detect subtle changes in the concentration of carbon dioxide (CO2) in their environment" And so what? What does CO2 means for insects? Why do they detect them? Where does it come from? Please give a wider introduction of this section.

About the conclusion: likewise. With "Our study will contribute to a better understanding of the important role that CO2 plays in the dispersal of H. cunea, which will provide a theoretical basis for pest manage-115 ment." It is not clear what this work adds to ongoing knowledge in the field of insect science/chemical sensing. How did the author do this study and for which reason they did so?

In addition, phrases like Heliconius Melpomene (H. melpomene) are not appropriate. It is sufficient to specify the full name, in this example Heliconius Melpomene, only once without specifications of its abbreviations in brackets, after which you can use the abbreviation H. melpomene in the rest of the text. I would accept so only if it is the journal to request it.

2.) At section 2-5 (line 260) the authors report p-values without specifying their statistical origins from the methods. Please provide statistics.

3.) This one of the most critical point that makes me pending between request of a further iteration and rejection: the new version of Figure 2 is not convincing. I still see no bootstraps at all on the one within the text. I tried to download the figure from the new zipped folder but it is still the old one. As I asked before, please replace this way to indicate bootstraps in numbers.

4.) This one of the most critical point that makes me pending between request of a further iteration and rejection: Figure 3: I do not understand what happened to Figure 3a, which is completely different from the previous one. 

5.) Still from Figure 3: I do not see a distinction between male and female body parts in qPCR, but reading the response to my comments, I understand the reason. I suggest the authors to discuss their motivation both in the method section at 4.5 and in the discussion as well including (when posssible) bibliographic references to similar cases to justify the reason of their chosing to do not distiguish these results between the two genders.

Minor issues

Line 177: write "body parts";

Line 199: just invert in the caption ELPG with EAG because ELPG is shown first in the figure;

Line 240: One question, looking at table S6: is the value of each biological replicate the average of its 3 technical replicates?  Please cite here Table S6. It is not cite elsewhere in the manuscript.

Line 300 and reference 28: You may have misunderstood my comment. Have a look at Clark 2019, "Leaves of terrestrial plants generally have more stomata on their undersides. Floating leaves of aquatic plants have stomata only on their upper surfaces, while some underwater plants have no stomata at all."

Ad this is obvious, to prevent plant dehydration. On a terrestrial plant leaf, upper side is wax mostly (cuticle) while the lower side has stomata for transpiration.  

Line 420: Please specify the use of F test to test variances and motivate your choice of using a homoscedastic two sample t test.

Line 460 See my comment in section 4.2.

Line 601: Please edit "Normorized" with "normalized" in table S6. 

FInd a pdf file attached.

Reviewer 3 Report

Comments and Suggestions for Authors

The authors have answered each of my comments, but answering the question does not always mean addressing the issue. 

My major concern is about the statistical analysis. It is very evident that it can and must improve. For example, the explanation of what consist a replicate in this study is hard to understand. In the first review I made several comments regarding this, but the answers are not satisfactory. 

I strongly recommend the authors to seek for advice from a statistician and perform the analysis properly, and to explain it clearly so we can assess if it has been done correctly.

In this vein, it would be very advisable that the authors provide the raw data of the electrophysiological and behavioral data in table format so that we can assess what they were analyzing.

One of the other reviewers had similar concerns and recommended rejecting the paper, so I feel that my claims are supported by other reviewers.

In an attached file I give my responses to the responses of the Authors, when applicable.

Comments on the Quality of English Language

Round 3

Reviewer 3 Report

Comments and Suggestions for Authors

BEHAVIOR

1-Atmospheric CO2 is 400 ppm or about 0.04% (https://news.climate.columbia.edu/2019/07/30/co2-drives-global-warming/)

2-The authors are using between 1 and 10%, which is about 1/0.04 and 10/0.04 times more than natural (25x and 250x). How realistic is this?

The choice test (Figure 1) is not analyzed statistically, but if it were and the data are what they appear to be by infering the values from the figure (they do not provide the actual values although they were requested to do so in the previous revision, precisely to check these aspects), a Fisher exact test would indicate that no concentration is significant in a two-tailed Fisher exact test (http://vassarstats.net/tab2x2.html)

10% c=20, co2=40, P=0.09

8% c=22, co2=38, P=0.19

5% c=24, co2=36, P=0.35

3% c=26, co2=34, P=0.58            

1% c=28, co2=32, P=0.85

Therefore, there is no behavioral effect of CO2

3-" Three biological repetitions were performed, and one biological replicate including 20 individuals". I am still unsure of what was the sample size. I am assuming 30 from looking at Figure 1, but I am not completely sure.

ELECTROPHYSIOLOGY

1- "A minimum of 3 individuals were tested and 3 technique replicates were performed for each individual". Why not just say "3 puffs"

2-It is very hard to do practically any statistics with N=3 (Lines 424-435). Mann-Whitney U test, maybe, but no t-test, please?

3-How was the CO2 "puffed"? How long were the puffs?

4-The authors were invited to fit non-linear dose response their EAG/ELPG data so that they could obtain E50 values and so compare their curves in a more elegant way than doing multiple statistical analyses for each concentration, which is what they (and many other researchers) do. The authors indicated that they could not fit curves to their data. Following is the fit of their data to non-linear dose response curves using de "drc" package and drm() function of R. I e with real data it may look different.xtracted the data visually from the Figure 3,

conc=c(1,3,5,8,10,1,3,5,8,10)

mV=c(6,7.5,8.5,12.5,12,4,4.5,5.5,6,6.5)

sex=c(rep("fem",5),rep("male",5))

model1=drm(mV~conc, sex, fct=LL.4(names = c("Slope", "Lower Limit", "Upper Limit", "ED50")))

plot(model1)

*****(see figure in attached file)

summary(model1) indicates that the ED50 of females and males is not that different: 5.15% and 4.94%, respectively. However the slope of the female curve is a lot larger than the slope of the male curve.

Obviously the female ELPG overall is a log higher in general than that of males (which could be tested with an overall t-test or ANOVA combining all the concentrations), but the dynamics of its response is very similar in terms of ED50 but very different in terms of slope (which is telling something about the "behavior" and the "quantity" of the receptors, respectively). Thus, exploring the curves in this way is a lot more informative than the multiple t-tests that the authors have performed.

It is notorious that the female ELPG is already kicking (7mV) at 1% CO2. This calls for a retest with lower percentages of CO2 (as indicated earlier the atmospheric CO2 is 0.04% and the authors are starting at 1%!!)

GENERAL

We still do not know at what time of the insect´s photoperiod were the behavioral tests performed

Comments on the Quality of English Language

must improve

Round 4

Reviewer 3 Report

Comments and Suggestions for Authors

See attached file (Review number 4)

Comments on the Quality of English Language

ok
